# A Predictive Multimodal Framework to Alert Caregivers of Problem Behaviors for Children with ASD (PreMAC)

**DOI:** 10.3390/s21020370

**Published:** 2021-01-07

**Authors:** Zhaobo K. Zheng, John E. Staubitz, Amy S. Weitlauf, Johanna Staubitz, Marney Pollack, Lauren Shibley, Michelle Hopton, William Martin, Amy Swanson, Pablo Juárez, Zachary E. Warren, Nilanjan Sarkar

**Affiliations:** 1Department of Mechanical Engineering, Vanderbilt University, Nashville, TN 37240, USA; nilanjan.sarkar@vanderbilt.edu; 2Treatment and Research Institute of Autism Spectrum Disorders, Vanderbilt University Medical Center, Nashville, TN 37240, USA; john.staubitz@vumc.org (J.E.S.); amy.s.weitlauf@vumc.org (A.S.W.); lauren.b.shibley@vumc.org (L.S.); michelle.hopton@vumc.org (M.H.); william.p.martin@vumc.org (W.M.); amy.r.swanson@vumc.org (A.S.); pablo.juarez@vumc.org (P.J.); zachary.e.warren@vanderbilt.edu (Z.E.W.); 3Department of Pediatrics, Vanderbilt University Medical Center, Nashville, TN 37240, USA; 4Department of Special Education, Vanderbilt University Medical Center, Nashville, TN 37240, USA; johanna.l.staubitz@Vanderbilt.Edu (J.S.); marney.s.pollack@vanderbilt.edu (M.P.); 5Department of Electrical Engineering and Computer Science, Vanderbilt University, Nashville, TN 37240, USA

**Keywords:** problem behaviors, wearable sensor, machine learning, affective computing, ASD, functional analysis, multimodal data, signal processing

## Abstract

Autism Spectrum Disorder (ASD) impacts 1 in 54 children in the US. Two-thirds of children with ASD display problem behavior. If a caregiver can predict that a child is likely to engage in problem behavior, they may be able to take action to minimize that risk. Although experts in Applied Behavior Analysis can offer caregivers recognition and remediation strategies, there are limitations to the extent to which human prediction of problem behavior is possible without the assistance of technology. In this paper, we propose a machine learning-based predictive framework, PreMAC, that uses multimodal signals from precursors of problem behaviors to alert caregivers of impending problem behavior for children with ASD. A multimodal data capture platform, M2P3, was designed to collect multimodal training data for PreMAC. The development of PreMAC integrated a rapid functional analysis, the interview-informed synthesized contingency analysis (IISCA), for collection of training data. A feasibility study with seven 4 to 15-year-old children with ASD was conducted to investigate the tolerability and feasibility of the M2P3 platform and the accuracy of PreMAC. Results indicate that the M2P3 platform was well tolerated by the children and PreMAC could predict precursors of problem behaviors with high prediction accuracies.

## 1. Introduction

Autism spectrum disorder (ASD) is a common neurodevelopmental disability characterized by social communication difficulties and repetitive patterns of interest and behaviors [1]. Current prevalence estimates indicate that one in 54 children in the US are diagnosed with ASD [2], two-thirds of whom display problem behaviors [3]. Although there are multiple terms that could be used for the very diverse behaviors targeted by our system, we utilize the term “problem behavior”, consistent with the Applied Behavior Analytic literature and the developers of the IISCA tool on which our system is based [4,5,6]. Common problem behaviors that co-occur with ASD include self-injury, aggression and elopement [7]. These behaviors severely impede involvement of children in community and educational activities [8] and can put children and their caregivers at risk of potential physical harm [9]. Persistent problem behaviors offer an important target for intervention because they can prevent children from learning new skills [10], excluding them from school services and community opportunities and aggravating financial burden on caregivers [11].

A validated practice for treating chronic problem behaviors is the Functional Analysis (FA), in which a Board Certified Behavior Analyst (BCBA) systematically manipulates environmental variables suspected to evoke and reinforce problem behaviors and directly observes the behaviors of concern under these controlled conditions in clinical settings in order to individualize treatment protocols that may benefit the child [12]. Although FA can provide an empirical understanding of the variables that impact behavior [13] and has been extensively researched, it is usually resource-intensive, requiring full engagement with a BCBA and other team members. In addition, while the significant resources invested in an FA may result in the identification of certain environmental variables likely to contribute to problem behavior, the FA stops short of building a model for truly predicting problem behavior outside of the clinical context. Disruptive, dangerous and chronic problem behaviors that occur outside of clinical settings and their corollary impact can lead to considerable stress for families, educators and children themselves, on top of the financial burdens of procuring best practice behavioral assessment and intervention services [14,15].

To address some of these limitations of the FA as most frequently described in the published literature, researcher-clinicians have developed a novel process for FA called the Practical Functional Assessment (PFA). The PFA leverages a structured interview with caregivers to identify the synthesized environmental variables most likely to evoke and reinforce problem behavior and then analyzes the occurrence of precursors to problem behavior when the synthesized contingencies described in the interview are systematically presented within the experimental design of the FA. The PFA has been studied as a means of increasing the safety, speed and acceptability of the FA process [16,17]. The PFA has demonstrated clinical utility when identifying and measuring precursor behaviors, which are observable behaviors—such as changes in body movement, affect or vocalizations—that reliably precede the onset of problem behaviors. In fact, it has been shown that precursors are functionally directly related to dangerous problem behaviors [18]. Because of this, assessors can use precursors as safe proxies for problem behaviors within the assessment context to reduce the potential for unsafe behavioral escalation.

The goal of the current work is to capitalize on the strengths of the PFA to develop a clinically-grounded multimodal data-driven machine learning (ML)-based problem behavior prediction model, PreMAC, which can be utilized within the community to potentially reduce the need for intensive human data collection. We hypothesize that with the advancement of wearable sensors and affective computing, it is possible to create a ML-based prediction model to accurately predict problem behavior (as well as observable precursors to problem behavior) using real-time sensor data that can provide minute changes in one’s internal and external states within a given context. 

Affective computing is an emerging field that aims to enable intelligent systems to recognize, infer and interpret human emotions and mental states [19]. There are many successful applications of affective computing to analyze and infer emotions and sentiments using facial expression, body gestures and physiological signals [20].

Affective computing has been successfully applied to inferring emotional and behavioral states of children with ASD based on various sensory data. Peripheral physiological responses such as heart rate (HR) and GSR have been used to predict imminent aggression [21]. The results demonstrated that the individualized and group models were able to predict the onset of aggression one minute before occurrence with good accuracy. With the same dataset, a more recent study [6] utilized support vector machine and it resulted in significantly better prediction accuracies over different prediction window lengths. In [22], skin conductance and respiration were used to build an ensemble of classifiers to differentiate the arousal level and valence in children with ASD. The results suggest the feasibility of objectively discerning affective states in children with ASD using physiological signals. With regard to behavior recognition from body motion, accelerometer data was used in [23] to recognize stereotypical hand flapping and body rocking behaviors, which may occur in some children with ASD. Stereotypical motor movements in ASD were detected using deep learning and resulted in a significant increase in classification performance relative to traditional classification methods [24].

In addition to the work on unimodal systems described above, several studies have shown promise regarding detection of affective and behavioral states of children with ASD using data from multimodal sources. For example, a multimodal stimulation and data capture system with a soft wearable tactile stimulator was developed to investigate the sensory trajectories of infants at high risk of ASD [25,26]. Wearable multimodal bio-sensing systems have been developed to capture eye gaze, EEG, GSR and photoplethysmogram (PPG) data [27]. Communication and coordination skills of children with ASD were assessed with multimodal signals, including speech, gestures and synchronized motion [28].

These and other existing studies demonstrate the potential of affective computing for children with ASD. With the advancement of low-cost robust sensors and computational frameworks it has become possible to create data-driven inference systems that are both accessible and affordable [29]. In general, multimodal systems that integrate several modalities, capture more information and hence increase the accuracy and robustness of machine learning models [30]. With regard to predicting precursors to problem behaviors, it is possible that including multiple modalities involving movements, physiology, social orientations and facial expressions could improve prediction accuracies and robustness. These modalities may directly capture the measurable indicators of emotional states of a child that may lead to problem behaviors such as fidgeting, arm crossing, cursing and grimacing [31]. Indeed, a recent study found that movement data along with annotated behaviors could build a machine learning model to predict episodes of SIB [32] but focused on prediction of problem behaviors themselves rather than precursors.

The primary contribution of the current work is the development of PreMAC that aims to predict imminent precursors of problem behaviors using multimodal data and behavioral states. Offering caregivers more time in advance could limit behavioral escalation and prevent dangerous problem behaviors. We present a novel PF- embedded experimental framework to collect training data for this model that seeks to capture expert BCBA’s direct behavior observations as the ground truth. In order to develop the PreMAC, we first created a novel Multimodal data capture Platform for Precursors of Problem behaviors, M2P3, for children with ASD. M2P3 combines an off-the-shelf wearable sensor, E4 [33], a Kinect sensor [34] and a customized Wearable Intelligent Non-invasive Gesture Sensor (WINGS). The presented multimodal platform is seamlessly integrated with a newly developed tablet-based software application, Behavior Data Collection Integrator (BDCI), to collect data and provide assistance to the assessment team completing a modified PFA. Note that the traditional behavioral assessment modalities rely primarily upon paper-and-pencil recording methods for data entry although there have been a few attempts recently to automate the process [35,36,37]. The customized BDCI help experts record ground truth for PreMAC in a convenient and precise manner that can be easily integrated with the M2P3- and WINGS-generated data.

The rest of this paper is organized as follows. Section 2 presents the overall framework to build PreMAC. Section 3 presents the details of the M2P3 platform design including sensor integration, software development and customized sensor design. Section 4 introduces the protocol of our feasibility study and pilot data collection. Section 5 presents the PreMAC training and prediction results. Finally, we conclude the paper with a discussion of results and potential future work in Section 6.

## 2. PreMAC Development Framework

The framework for the development of PreMAC is shown in Figure 1. We have embedded the Interview Informed Synthesized Contingency Analysis (IISCA) into our data collection for training the prediction model. IISCA is a commonly used type of PFA [38]. In an IISCA assessment, a BCBA methodically manipulates the environment to test caregiver hypotheses around what environmental stimuli serve as antecedents or establishing operations (EOs) to a problem behavior, and which stimuli serve as reinforcers to the problem behavior. During the IISCA, the child interacts with the BCBA who systematically evokes problem behavior (or more often the reported precursors to problem behavior) by presenting the EOs identified by the caregiver. The BCBA then provides contingent reinforcement for the problem behavior or precursor to halt behavioral escalation and verify that the child’s behavior is functioning to receive the reinforcers indicated by their caregivers. At the same time, M2P3 is deployed for use with the children with ASD for multimodal data collection. Another BCBA observer watches the sessions and uses the BDCI to report their ratings on the behaviors of the child. The multimodal data are then denoised, synchronized and processed and are used to extract features. The ratings of the observer for the behavioral states are used as the ground truth. The features of the multimodal data are then mapped against the ground truth to train PreMAC to predict the precursors of problem behaviors. Cross validation is then run on the PreMAC to address its accuracy and analysis results.

## 3. Multimodal Data Collection Platform Design

In order to collect adequate multimodal signals for PreMAC, we developed the M2P3. It integrates and synchronizes multiple data modalities of different time scales. The platform architecture is shown in Figure 2. The data modalities of M2P3 include facial expressions and head rotations from the Kinect, peripheral physiological and acceleration signals from the E4 and body movements from WINGS. We also developed a tablet application, BDCI, to collect direct behavior observation data.

### 3.1. Kinect and E4 Sensors

M2P3 consists of several platform components. A Microsoft Kinect V2 was used to detect the facial expressions and head rotations of the children. Microsoft Kinect API computes positions of eyes, nose and mouth among different points on the face from its color camera and depth sensor to recognize facial expressions and compute head rotations. We integrated the API to read these measurements in C# scripts. M2P3 is designed to track the first child that enters the camera view of the Kinect. The facial expressions that can be recognized by the API are: happy, eyes closed, mouth open, looking away and engaged. These measures are classified with facial features in real-time and vary on a discrete numerical scale that ranges from 0, 0.5 and 1, meaning no, probably and yes, respectively. Facial expressions such as happy and engaged are not determinant measures of arousal but have strong indicators of such states [39]. Whether the child is engaged is decided by whether the child opens both eyes and look towards the Kinect. The head rotations are measured in terms of roll, pitch and yaw angles of the head. The sampling rates of the head rotations and facial expressions are both 10 Hz and the signals are recorded with time stamps with millisecond precision. The Kinect is placed on the wall by a 3D printed structure which can adjust the pan and tilt angles of the Kinect so that it directly faces the child as shown in Figure 3.

Four physiological signals—blood volume pulse (BVP), electrodermal activity (EDA), body temperature and three axis acceleration from an accelerometer—are collected through the E4 wristband. The wristband itself is noninvasive and resembles a smart watch. The sampling rates for BVP and EDA are 64 Hz and 4 Hz, respectively. We used the API provided for the E4 to record the data with precise time stamps. The real-time physiological data stream is transferred to a central controller by wireless Bluetooth communication.

A central controller is created in Unity, a widely used game engine [40], in C#, to integrate all the data collection modalities. The data collection can be started or stopped by the click of a button. The user interface also displays data being captured by the console and a point cloud showing the field of view of the Kinect.

### 3.2. WINGS

The Wearable Intelligent Non-invasive Gesture Sensor or WINGS is a body movement tracking sensor designed for children with ASD. It is a portable, noninvasive tool for measuring upper body motion as shown in Figure 4a,b. There are, in general, two popular ways to track motion and gestures: one is based on computer vision (CV) and the other is based on inertial measurement units (IMU) [41,42]. Despite CV being less-invasive, it has limitations with regard to field of view, occlusion, portability and computational demands [43]. On the other hand, the IMU-based gesture sensor although body worn, could be a better solution in unstructured environment such as in homes and schools where the children will move around. WINGS integrates IMUs to measure the acceleration and orientation of the torso and limbs using a combination of accelerometers and gyroscopes. To increase the likelihood that the platform will be tolerated by children with varying levels of activity, sensory sensitivity and cognitive functioning, we created WINGS within an off-the-shelf cotton hoodie where the IMUs [44] are sewn within an enclosed space between inner and outer cloth layers. The remaining electronic components including controllers, battery, transmitters and the circuit are sewn within the hood.

Children cannot see or touch any of the electrical and mechanical elements. The total weight of WINGS is 232 g. When worn, it feels like a normal hoodie. WINGS presents the advantage of allowing children to have an unrestricted workspace. However, we note that some children with ASD will not tolerate wearable sensors and in such cases, WINGS will not be the solution. The total cost of one WINGS is about 170 dollars, although the unit cost will reduce as the production increases. A variety of sizes of WINGS were made to fit children of different sizes.

The electronic components of WINGS include an Arduino Uno microcontroller, an I2C multiplexer, a 9 V battery, a wireless transmitter and 7 IMUs. Figure 4c shows the data flow scheme of the system. In order to fully construct the upper-body gestures of a child wearing WINGS, we need 7 IMUs to measure joint angles of each forearm, upper arm and the three locations on the back for optimal sensor locations for self-stimulatory behaviors detection [45]. Four cables from each IMU connect to the Uno controller hidden in the hood. Each IMU uses an I2C communication with the Uno microcontroller while the I2C multiplexer [46] searches and loops through the IMUs. The Uno sends the data via a wireless transmitter to a 2.4 GHz receiver and the receiver then sends the data further to an Arduino Mega microcontroller. The wireless transmitter and receiver have a SPI communication with the Arduinos. The Mega controller sends the data to a workstation for data storage through a serial communication. When tested, the battery life for WINGS was more than 25 h, which is adequate for sessions in clinic, school and other outpatient settings.

From the 3 components of the accelerometer readings, ${\mathrm{accl}}_{x}$, ${\mathrm{accl}}_{y}$ and ${\mathrm{accl}}_{z}$ and 3 components of the magnetometer readings, ${\mathrm{mag}}_{x}$, ${\mathrm{mag}}_{y}$ and ${\mathrm{mag}}_{z}$, we can compute the roll, pitch and yaw angles (θ, ψ, ϕ) of the torso and limbs using Equations (1)–(3) as shown below. The roll and pitch angles are computed by the IMU orientations with respect to the gravitational direction. The yaw angle is computed by the relative IMU orientations with respect to the earth’s magnetic field.
(1)$${\theta =\tan}^{-1}\left(\frac{{\mathrm{accl}}_{y}}{\sqrt{{\mathrm{accl}}_{y}^{2}{+\mathrm{accl}}_{z}^{2}}}\right)$$
(2)$${\psi =\tan}^{-1}\left(\frac{{\mathrm{accl}}_{x}}{\sqrt{{\mathrm{accl}}_{y}^{2}{+\mathrm{accl}}_{z}^{2}}}\right)$$
(3)$$\phi {=\tan}^{-1}\left(\frac{{\mathrm{mag}}_{z}{s\psi -\mathrm{mag}}_{y}c\theta }{{\mathrm{mag}}_{x}{c\theta +\mathrm{mag}}_{y}{s\theta s\psi +\mathrm{mag}}_{z}c\psi s\theta }\right)$$

Knowing the roll, pitch and yaw angles of different joints, we are able to compute the 3D positions and orientations of each joint using forward kinematics [47]. As shown in Figure 5a, the base frame is set at the spine base of the child. The base frame’s positive directions along the x, y and z axes are front, left of the child and up, respectively. Then the coordinate frame $s_{n}$ is attached to each body joint. Homogeneous transformation matrices $H_{\mathrm{Jointn}}^{\mathrm{Jointn}-1}$ between the *n*th joint and the (n − 1)th joint consist of two parts: a 3-by-3 rotation matrix $R_{n}^{n-1}$ and a 1-by-3 translation vector $d_{0}^{n-1}$. The rotation and translation matrices can align and move the previous coordinate frame to the current coordinate frame, respectively. The rotation matrix is computed by roll, pitch and yaw angles while the translation vector is computed by the body link lengths which are manually measured for different sizes of WINGS. Each homogeneous transformation matrix is computed using Equation (4).
(4)$$\begin{array}{c} H_{\mathrm{Jointn}}^{\mathrm{Jointn}-1}=\left[\begin{array}{cc} R_{n}^{n-1} & d_{0}^{n-1}\\ \vec{0} & 1 \end{array}\right]=\left[\begin{array}{cc} R_{x,\psi }R_{y,\theta }R_{z,f} & d_{0}^{n-1}\\ \vec{0} & 1 \end{array}\right]=\\ \left[\begin{array}{cccc} {\mathrm{cf}}_{n}{c\theta }_{n} & {\mathrm{cf}}_{n}{s\theta }_{n}{s\psi }_{n}{-\mathrm{sf}}_{n}{c\psi }_{n} & {\mathrm{sf}}_{n}{s\psi }_{n}{+\mathrm{cf}}_{n}{s\theta }_{n}{c\psi }_{n} & x_{n}^{n-1}\\ {\mathrm{sf}}_{n}{c\theta }_{n} & {\mathrm{sf}}_{n}{s\theta }_{n}{s\psi }_{n}{+\mathrm{cf}}_{n}{c\psi }_{n} & {\mathrm{sf}}_{n}{s\theta }_{n}{c\psi }_{n}{-\mathrm{cf}}_{n}{s\psi }_{n} & y_{n}^{n-1}\\ {-s\theta }_{n} & {c\theta }_{n}{s\psi }_{n} & {c\theta }_{n}{c\psi }_{n} & z_{n}^{n-1}\\ 0 & 0 & 0 & 1 \end{array}\right] \end{array}$$

The overall homogeneous transformation matrix $H_{\mathrm{Jointn}}^{\mathrm{Origin}}$ between the base frame and the *n*th frame can be computed by multiplying all the homogeneous transformation matrices as in Equation (5). From this matrix, $d_{n}^{0}$ provides the 3D position of the *n*th joint position with respect to the base frame.
(5)$$\begin{array}{c} H_{\mathrm{Jointn}}^{\mathrm{Origin}}{=H}_{\mathrm{Joint}1}^{\mathrm{Origin}}{\mathrm{gH}}_{\mathrm{Joint}2}^{\mathrm{Joint}1}{\mathrm{LH}}_{\mathrm{Jointn}}^{\mathrm{Jointn}-1}\\ =\left[\begin{array}{cc} R_{1}^{0} & d_{1}^{0}\\ \vec{0} & 1 \end{array}\right]L\left[\begin{array}{cc} R_{n}^{n-1} & d_{n}^{n-1}\\ \vec{0} & 1 \end{array}\right]=\left[\begin{array}{cc} R_{n}^{0} & d_{n}^{0}\\ \vec{0} & 1 \end{array}\right] \end{array}$$

Thus, we have the 3D positions of each body joint and we can construct the body gestures made using these joints. A MATLAB program was written to visualize the upper body gestures in real time. Figure 5b shows a visualized gesture and Figure 5c shows its corresponding photo. The lines represent the limbs and the blue dots represent the joints.

The precision of the IMU measured roll, pitch and yaw angles is approximately 1 degree. To quantitatively validate the overall precision of WINGS, we conducted a test where a user wore WINGS and sat in a chair at a designated point. Then the user reached nearby designated 3D points using his shoulder, elbow and wrist. Thus, the relative 3D positions between that joint and the spine base could be measured manually and we compared it to the results computed by WINGS. The user used each joint to reach the designated point for 10 times and the average errors of the shoulder, elbow and wrist were 5.7 mm, 9.6 mm and 11.7 mm, respectively. These precisions are adequate for human gesture measurements for our purpose.

### 3.3. Behavioral Data Collection Integrator

To record under which conditions target behaviors were observed, the IISCA requires observers to record the occurrence of precursors to problem behaviors or problem behaviors themselves, typically using paper and pen while timestamping events via a stopwatch [48]. There have been some attempts recently to automate this process. A computerized behavioral data program “BDataPro” allows real-time data collection of multiple frequency and duration-based behaviors [35]. Catalyst, another software for behavioral assessment, allows collection and management of a wide variety of data for behavioral intervention, including skill acquisition and behavior reduction [36]. An annotation tool for problem behaviors for people with ASD was also developed to log data more conveniently [37]. These existing annotation tools cannot efficiently and precisely record and integrate direct behavioral observation with multimodal data collection. To increase the portability, convenience and precision of behavioral data collection, we designed a tablet application to assist human therapists with recording data during IISCA procedures, the BDCI. BDCI was written in Unity and implemented on an Android tablet [49].

The application has three pages: Initialization, Session and Summary. In the Initialization page, there are fields for the observer to input child information, therapist information, as well as session number and type. Once initialized, the observer clicks the start button to begin the session. In the meantime, the application generates a text file to store information and the interface moves to the second page, the Session page. By clicking each button, the application writes a data entry containing the category of the event and its time stamp precise in milliseconds. As shown in Figure 6, there are several buttons on the Session page related to observer actions and child behaviors. Two buttons are available for the observer to switch between two therapist-imposed conditions within this assessment protocol: establishing operations (EO) and Reinforcing stimulus (S^R^). Establishing operations represents those antecedent conditions reported to evoke behavioral escalation by caregivers. Reinforcing stimulus (S^R^) represents those intervals in which antecedent conditions are arranged to prevent, de-escalate and restore a state of happy and relaxed engagement.

For this app, the current antecedent condition is highlighted in green. The observer can toggle between the two conditions by clicking the relevant button. Event recording of problem and precursor behaviors are recorded by the app. The elapsed time for the current assessment session and condition within the session are shown in the lower part of the screen. According to the IISCA protocol, a specified minimum duration of 90 s of the child demonstrating a happy, relaxed and engaged affect within the S^R^ condition is needed prior to re-instituting the EO condition. This procedure is used in order to prevent the child from escalating to higher intensities of problem behavior or becoming emotionally dysregulated to such a degree that there is a reduction to their awareness of their environment. This second point may sound counter-intuitive as therapists are teaching the child to engage in high rates of undesirable behavior, but bear in mind that it is the earliest and least disruptive form of the child’s escalation cycle that is being strengthened through this assessment, and it is when high rates of precursor behavior are evoked within the experiment that a robust individualized prediction model for problem behavior can be built. The app includes stopwatches for time management that can cue the data collector and BCBA when a change of condition is appropriate. When an antecedent condition is not ready to be implemented, the button turns red and includes a countdown for the time remaining until the next condition can be implemented.

The app was designed using a finite state machine (FSM) that integrated with our modified IISCA protocol. The app starts with the initialization state. After logging in the session information, the app goes to the S^R^ state. It was important to the assessment process and our subsequent analyses to include time stamps for when the child was demonstrating a calm affect. At the outset of each experiment, therapists discussed the importance of collecting this information with caregivers and sought their assistance in using their expert knowledge of their child’s affective states to ensure that the data collector was accurate in recording periods of observable calm in the participating child. Caregivers observed every minute of every experiment through a one-way mirror and provided real-time feedback to the data collector as to when the child became or ceased to be calm. The data collector in turn pressed the calm button within the BDCI app and a timer provided feedback to the data collector as to the duration of the current interval of calm. A continuous happy, relaxed and engaged state lasting at least 90 s was sought (by keeping reinforcement in place) to prevent behavioral escalation and give the child’s body time to provide “calm” data to the M2P3 and WINGS that could be compared with the data generated when they were escalating behaviorally. If any precursor or problem behaviors happen during this time, the S^R^ condition must continue. If the child is observed to remain continuously calm, the app indicates a readiness for the EO conditions at the end of 90 calm seconds and the observer will click the EO button as the therapist begins to present the evocative EO conditions. In the EO state, if the precursor button is clicked, the event is recorded, and the app will provide a 90 s count down after which the app indicates readiness for the S^R^ condition. If a single assessment session is finished, the app proceeds to the summary state; if all the sessions are already finished in the summary state, the app will move to the end state. The FSM is shown in Figure 7. BDCI provides better precision of behavioral data collection and it is deployable on Android, IOS and Windows platforms. Given the ubiquitous nature of these devices, we anticipate very low-cost burden; indeed, it may reduce cost by eliminating training of collecting observational data and the necessity of including multiple observers for interobserver agreement.

## 4. Data Collection Experiment

In order to collect training data for PreMAC and also to demonstrate the feasibility and tolerability of M2P3, we conducted a feasibility study with 7 participants with ASD from 4 to 15 years old (6 male, 1 female; mean age = 10.71 years, SD = 3.1472). These children all had diagnoses of ASD from licensed clinical psychologists. Participants’ caregivers reported that the participants presented with frequent episodes of problem behavior which are predictable and significant enough to be provoked by a novel therapist within a novel clinical setting as part of the study protocol. The protocol was reviewed and approved by the Institutional Review Board (IRB) at Vanderbilt University. The research team members explained the purpose, protocols and any potential risks of the experiment to both the parents and the participants and answered all their questions before seeking informed consent from the parents and informed assents from the participants. Because the purpose of the study was to evoke and respond to precursors to problem behaviors and prevent escalation to dangerous problem behaviors, parents and two dedicated BCBA data collector observed the assessment sessions to ensure that all precursors and problem behaviors as well as the target emotions of happy, relaxed and engaged were correctly recorded. Behavioral states were coded accordingly to clearly-defined written criteria across two observers, as described above. The precursors and problem behavior episodes and the calm states were noted by the observers with the help of observing caregivers and then recorded by the observers using BDCI.

### 4.1. Experimental Setup

As shown in Figure 8a, the child-proof room has two compartments, the experimental space and the observation space. The participant sits in the experimental space with a BCBA therapist. The seat for participants is 2 m away from the Kinect and a video camera. The participant wears an E4 sensor on the nondominant wrist and WINGS on the upper body. Four observers including an engineer, one of the participants’ primary caregivers, a BCBA data collector and a BCBA assessment manager are seated in the observation space, which has a one-way mirror towards the experimental space. The observers and the parent can see the therapist and the participant through a one-way mirror. The therapist had a Bluetooth headphone to relay information from the manager and the manager ensured that the time components of the experimental protocol were correctly executed.

The participant was first invited to the experimental space by the therapist. Then the door was closed to separate the experimental space from the observation room. The therapist then put the E4 sensor on the wrist of the participant and helped him or her wear WINGS. Meanwhile, the parent and the other observers entered the observation room. The Kinect can track up to 16 people at the same time by assigning a specific body ID for each user. In this experiment, the Kinect calibration was performed with only the participant in the Kinect camera view. In this way, the body ID of the participant was recognized so the program only recorded the data of the participant and not the therapist. Each experiment lasted for approximately one hour.

### 4.2. Experimental Procedure

The experiment followed a modified IISCA protocol [50]. We conducted multiple therapeutic sessions in a single experimental visit to capture data on different behavioral states. These sessions are labeled as control (C) and test (T). The sessions are structured as CTCTT, which represents a multielement design for single subject research [51]. The control sessions contain only S^R^ conditions and the test sessions alternate between EO and S^R^ presentations. EO is followed by S^R^ and EO is applied once again after at least 90 s have elapsed during which the participants stay calm. During EO presentations, the therapist simulated the antecedent conditions that were most likely to evoke precursors and problem behaviors. These tasks were reported by the parents in an open-ended interview days before the actual experimental visit. The most commonly reported tasks that induced problem behaviors include asking them to complete homework assignments, removing preferred toys or electronics from them and withdrawing preferred social attention from them. During S^R^ condition presentations, the therapist offers free access to their favorite toys and electronics, stops asking them to work, removes all the work-related materials and provides them with the reported preferred attention such as making eye contact, smiling and showing interest. The primary caregiver of the participants observed from behind the one-way mirror, watched the behaviors of the participant and gave feedback to the data collector and manager who verified the occurrence of precursors or problem behaviors and the presence or absence of a calm state. At times, the caregiver provided advice on how to calm the child or how to provoke problem behavior. The structure of the whole experimental procedure is shown in Figure 8b.

### 4.3. Feasibility Study Results

All 7 participants completed their entire experimental visits. The average length of session time was 54.2 min (min = 36.5 min, max = 63.1 min, SD = 11.5 min). The average duration for each session is 12.05, 11.37, 10.02, 10.71 and 10.05 min, respectively. The time variation across sessions was largely due to differences in how long it took for each participant to calm down during S^R^ sessions. The average of precursors observed was 25.9 episodes (min = 21, max = 30, SD = 3.02). WINGS was the most invasive component in the M2P3 platform; 6 out of 7 participants tolerated it without a problem. Some participants even put WINGS on themselves. The only participant who did not tolerate WINGS the entire time put it on at the beginning and then decided to take it off after 15 min because he had a high level of caregiver-reported tactile sensitivity.

The other wearable platform component, the E4 wristband, was less invasive and tolerated well by all participants. With regard to staying within the view of the Kinect, one participant was unable to stay seated at the table throughout the entire experiment and instead spent some time on the floor with toys. Thus, the Kinect was not able to track the participant for the entire duration of the experiment.

## 5. PreMAC Training

### 5.1. Multimodal Data Collection and Signal Processing

M2P3 collected data using four components: WINGS, E4 wrist band, Kinect and BDCI. WINGS provided movement data of the upper body; E4 sensor provided peripheral physiological data and the 3-axis acceleration signal of the wrist; Kinect provided facial expressions and head rotations data; and BDCI supplied behavioral states.

Movement data collected by WINGS had a sampling rate of 15 Hz. Less than 0.3% of WINGS data entries were corrupted due to wireless communication or signal noises. A low-pass filter was applied with a cut-off frequency of 10 Hz to raw signals for accelerations that contained sensing noises. The threshold was chosen according to the usual speed of human motions [52] so that the noises were filtered out while keeping information-rich signals for analysis. Peripheral physiological data, BVP and EDA, were collected with the sampling rates of 64 Hz and 4 Hz, respectively. BVP signals were filtered by a 1 Hz to 8 Hz band pass filter. Acceleration signals of the E4 were collected at 32 Hz and a low-pass filter of 10 Hz was applied to it.

There was a significant amount of missing data for the head rotations and facial expressions. The Kinect failed to collect these measurements 26.9% of the time when participants were looking down or away. An interpolation algorithm was used for the missing data points, where the numerical mean value of the 20 closest available head rotations and the most frequent class among the 20 closest available facial expressions were chosen respectively, to fill the missing data.

### 5.2. Feature Extraction

From the processed and filtered data, different features were selected and extracted based on published literature on problem behaviors and stress detection as well as the insight from the BCBAs involved in our study. Certain gestures, including head banging, fist throwing, body swinging and repetitive arm movements are strongly related to problem behaviors [53,54]. Since the roll, pitch and yaw angles of various joints in the body are able to indicate physical movements of orientation. The rotation angles of the head, the torso and the limbs were extracted as features from both the Kinect API and WINGs signals. These features can construct the upper body gesture using Equations (1)–(3). Besides gestures, the average magnitude of accelerations shows the physical movement intensity of the children. Fast and repetitive movements such as fidgeting are a common category of problem behaviors [55], where increased activity levels take place. Facial expressions, such as pouting, eyebrow knitting and intensely staring are observable precursors. From the Kinect API, we extracted features of closing of eyes and mouths, engagement and looking away. We analyzed the predictive power of these extracted features by feature importance, presented later.
(6)$$\mathrm{AL}=\frac{\sum_{i=1}^{n}\sqrt{a_{x_{i}}^{2}{+a}_{y_{i}}^{2}{+a}_{z_{i}}^{2}}}{7}$$

Peripheral physiological data is a strong indicator of arousal and several features including heart rate (HR) and EDA have been shown to have correlations with problem behaviors [21,22]. Thus, the HR level was computed by interbeat-intervals (IBI) of the BVP raw signal. From the EDA data, two types of data were separated and characterized, which were tonic skin conductance level (SCL) and phasic skin conductance response (SCR) [56]. These features correlate well with the arousal level of a person [57].

Features from different modalities were combined for training and testing of PreMAC. There were altogether 32 features: 20 roll, pitch, yaw and activity level from forearms, upper arms and torso from WINGS; 1 heart rate and 2 skin conductance features from E4; 3 accelerations from E4, and 6 facial expressions from Kinect. These features were generated at different instants due to varying processing times of different signal modalities. In this work, in order to combine all these features for each time step, we used WINGS’ features at any given instant as the basis and added the other features mentioned above that were closest in time with the WINGS’ features.

The BDCI provides the time stamps of precursors of problem behaviors captured by the BCBA observers. With these time stamps, we assigned either absence or presence of imminent precursor classes to each 1-by-27 vector of multimodal data. With the insight from our IISCA practitioners, we chose the most representative data for precursors of problem behaviors within a time window. The time window is between 90 s before the episode and the point of precursor generation. Thus, if the multimodal dataset was collected within 90 s prior to the precursor, it would be assigned label 1. Otherwise, the class was assigned label 0. The two classes 0 and 1 had an average ratio of 6:4. This was not a significantly unbalanced dataset so class balancing was not necessary to maintain more information. For our experiment, each child had an average of 27,242 samples of the multimodal data.

### 5.3. Machine Learning

PreMAC includes both individualized models and group models, predicting whether a precursor to problem behaviors is going to happen in a window of time. The individualized models were built with data from each participant. The group models were built with data from all the participants to explore the general group behavioral patterns. In order to find the most accurate ML algorithm, we explored several standard ML algorithms with our datasets. The library scikit-learn [58] was used on Jupyter Notebook [59]. For individualized models, the samples were randomly divided into training and test sets with a ratio of 80 to 20. Then a 5-fold cross validation was run to compute the accuracies of each algorithm. For the group models, we chose a leave-one-out validation, which uses a selected participant’s data as the test data to determine prediction accuracy based on training utilizing the rest of the data. The prediction accuracies of models based on several algorithms are shown in the table below.

Table 1 shows the accuracies of different algorithms and the individualized model accuracies are the average accuracies among all the participants. For individualized models, Random Forest (RF), k Nearest Neighbors (kNN), Decision Tree (DT) and Neural Network (NN) have high prediction accuracies while Support Vector Machine (SVM), Discriminant Analysis (DA) and Naïve Bayes (NB) have comparatively lower accuracies. The RF classifier had the best prediction accuracy potentially due to its nature of uncorrelated decision trees operating as a committee, with bagging to prevent overfitting [60]. It also works well with the high dimension of features and nonlinear data, making it a very popular classifier for remote sensing community [61]. For group models, RF and NN show good accuracies while SVM, kNN and DT have significant decrease in prediction accuracies. The group model of DA and NB perform very poorly, close to random guessing. More importantly, we observe a significant drop of prediction accuracies for group models despite the fact that the combined samples from all participants could help make more accurate prediction. The leave-one-out validation of the group model is using data of other children to predict the behaviors of a new child. The results indicate that the personal behavioral patterns related to imminent precursors vary significantly among children and a group model trying to predict the average problem behaviors could not achieve outstanding accuracy in this work. The RF individualized model has the best average prediction accuracy. The confusion matrix of an example individualized model is shown in Table 2. The confusion matrix of the group model is shown in Table 3.

The RF algorithm also offers estimates of importance of each feature. The feature importance is computed as the total decrease in node impurity weighted by the probability of reaching that node. We also analyzed the relative importance of motion-based, physiology-based and facial expression-based features. The features included were motion signals of each joint, physiological signals, head rotations and facial expressions. As shown in Figure 9c, the most important features were from torso, right shoulder and right wrist, where the total of all features equals to 1. The results are consistent with our experimental observations. The main precursors included participants banging their arms against their torsos and moving their right arms, as the right arm was the dominant arm for the participants. For data modalities from E4 sensor, the physiological features including both HR and EDA had an importance of 0.0566 and the 3-axis accelerations had an importance of 0.0689. Head Rotations had an importance of 0.0399 and facial expression features were the least important with a value of 0.0061. The poor performance of facial expressions may be due to the missing portion of data when children were looking down and away.

We also analyzed the prediction accuracies with only WINGS data, only physiological data and only E4 data to compare the contributions of different data modalities. We utilized the best performing algorithm RF to learn the data pattern of each child. The average prediction accuracies and the range of each model are shown in Figure 9a. The average prediction accuracies for the multimodal model, physiological data only model, WINGS data only model and E4 data only model were 98.51%, 85.22%, 87.31% and 91.63%, respectively. Multimodal individualized models have high accuracies with small variances, meaning the performance is robust among children. The customized WINGS provides data for significantly improved prediction accuracies compared to the commercial E4 sensor.

As mentioned earlier, label 1 (imminent precursor) was assigned to data that was collected within the last 90 s prior to the observation. To analyze the effect of the time window on the prediction of precursors, we varied the time window for the class of imminent precursors from 30 s prior to the observed precursor to 120 s in steps of 30 s. To avoid effects of different ratio of classes, we oversampled the minority class so that the two classes had a 1:1 ratio. As shown in Figure 9b, prediction accuracies for 30, 60, 90 s do not have significant differences but it significantly decreases for the 120 s time window. It is noteworthy that this reliable 90 s prediction is for the precursor that reportedly precedes actual problem behavior by an unknown number of seconds or minutes. A 90-s warning prior to even precursors happening provides enough time for a caregiver to withdraw a demand, provide desired attention, or redirect the child to a preferred item or activity. This analysis validates that the 90 s window seems to be the optimal window for precursor prediction that has good prediction accuracy with ample time for the caregivers to intervene.

## 6. Conclusions

Best practice models for assessing problem behavior in order to inform interventions for preventing de-escalating or teaching alternative behaviors for children with ASD currently do not provide a real-time prediction model. To augment and extend a best-practice clinical assessment model, which is necessary for individualizing intervention approaches for individuals with ASD and other developmental disabilities [16], we developed a novel machine learning based predictive model, PreMAC. Based on multimodal data input, PreMAC creates individualized and group profiles of imminent behavioral escalation among children with ASD based upon physiological, gestural and motion-based precursors that a problem behavior is about to occur. This multimodal data capture platform, M2P3, collects training data from two portable wearable devices (including one of our own creation, WINGS) and a newly designed tablet application, BDCI, all of which represent low cost options for future real-world community deployment.

PreMAC integrates important relevant data that cannot be reliably collected by a human observer. Specifically, it collects data regarding only subtly visible (e.g., joint angle) or utterly invisible (e.g., skin conductance) precursors of problem behavior at a high level of accuracy. The emphasis of our system design on precursors rather than problem behaviors themselves holds the potential to increase the safety of participants by minimizing the risk of a severe problem behavior actually occurring during the sessions needed to build the predictive model. In summary, this system rapidly generates a robust prediction model with ample time to be clinically and practically relevant all with little-to-no dangerous behaviors occurring at any time during the assessment. If integrated within a system that could somehow signal an adult, this would give caregivers and potentially people with ASD themselves more lead time prevent or quickly de-escalate problem behaviors. When reactive procedures are needed to protect the child or caregiver, advance notice of 30–90 s can make a difference in safety. Additionally, a reliable prediction model could be leveraged to improve intensive intervention procedures, enhance staff and caregiver training and improve fidelity to plans for preventing and reacting to problem behavior.

Our innovative data collection process is novel in its integration of multimodal data collection with cutting edge functional assessment technology from the field of Applied Behavior Analysis. Each step of this work was informed by stakeholder feedback which was then integrated into the system design. Importantly, particularly when designing a system intended for future real-world clinical use, results of this feasibility study suggest that children with ASD with problem behaviors tolerated both the platform and experimental protocol well. The protocol also efficiently evoked and reinforced precursors in participating children without the occurrence of dangerous or disruptive problem behavior or emotional responding, an outcome likely to promote caregiver acceptability.

To our knowledge, PreMAC extends existing sensory modalities of problem behavior prediction with upper body motion and social orientations and it is the first machine learning model to integrate an IISCA to evoke precursors to problem behaviors instead of dangerous episodes of problem behavior. Within our controlled laboratory context, PreMAC offered a significant increase in prediction accuracy, an average of 98.51% for individualized profiles, as compared to the existing published results which predicted behaviors themselves rather than precursors [6,23,32]. Potential reasons for higher prediction accuracy of PreMAC include more sensing modalities, more accurate precursor time stamps through BDCI and large data sample size of each child. It is also worth mentioning that this work is predicting behavioral precursors of problem behaviors that precede problem behavior episodes, demonstrating great potential to offer more time in advance for caregivers to intervene.

Based on our analysis, body motion is the most predictive sensing modality for imminent precursors of problem behaviors and WINGS alone may provide adequate information to predict imminent precursors. We further investigated the importance of different limb movements, head rotations, physiological data and facial expressions. The torso movement is the most effective feature and movement on the dominant side is more effective than the other side. Physiological data is comparatively much less effective than body movements and facial expressions almost do not contribute to the prediction accuracies. This paves the way for future work to identify the most efficient sensor to integrate into an online platform for home and school settings.

Several limitations exist that warrant attention in future work. First, the Kinect and the video camera are the two nonportable components in the platform that, at present, impede data collection in an out-of-lab setting. In the future, we will continue working towards a totally portable data collection platform for home and school settings, which will better assess behaviors of children with ASD and usual problem behaviors. WINGS have combined upper body motion detection with the softest clothing most typically worn by children in this age group and further testing will include more stakeholder input including questions about possible improvements to increase maximum comfort level across a broad range of sensory profiles. Because this is not an autism-specific system, but rather one designed for any child with problem behaviors, updated phenotypic information was not obtained for the purpose of this small pilot study. The group model is not a great predictor of individual precursors, at least based on this small sample study. There is also no guarantee in our study design that precursors will be generated. In future work, we will obtain measures of autism severity, problem behavior frequency and cognitive skills using standardized tools to better understand the likely variability that will present across a larger sample of individuals. We will also evaluate the functions of the system if precursor behaviors do not occur. In that way, the system will be able to catch physiological precursors that are not observable by human. In spite of these limitations, the proposed platform collects multimodal data with wearable sensors including customized WINGS, a novel tablet application gathering precise time stamps for function analysis and an IISCA protocol to generate high-density precursors with very few actual problem behaviors. The platform was validated on seven children with ASD and the performance of PreMAC was promising.

## Figures and Tables

**Figure 1 sensors-21-00370-f001:**
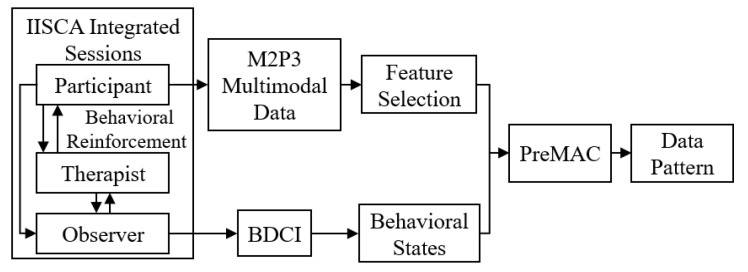
Study flow chart.

**Figure 2 sensors-21-00370-f002:**
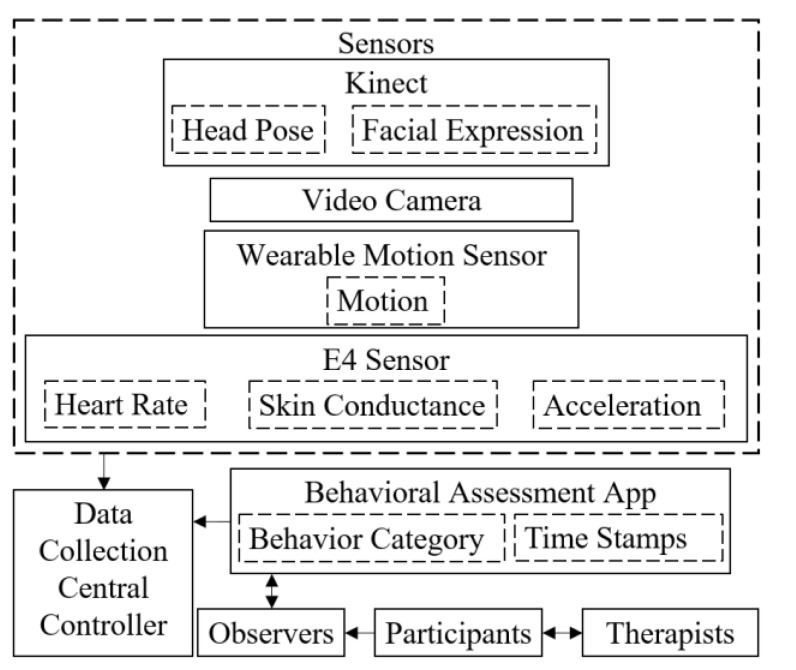
Platform architecture.

**Figure 3 sensors-21-00370-f003:**
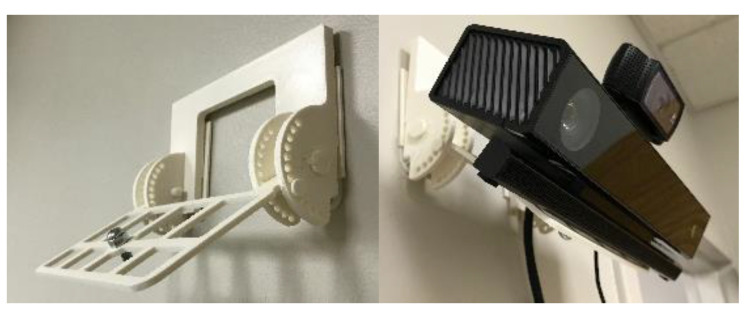
Kinect setup.

**Figure 4 sensors-21-00370-f004:**
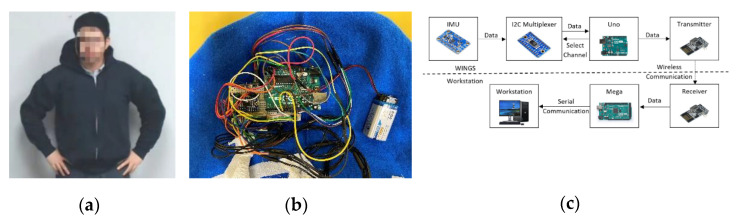
(**a**) Wearable Intelligent Non-invasive Gesture Sensor (WINGS) on a user; (**b**) Electronics hidden inside WINGS; and (**c**) WINGS electronic design.

**Figure 5 sensors-21-00370-f005:**
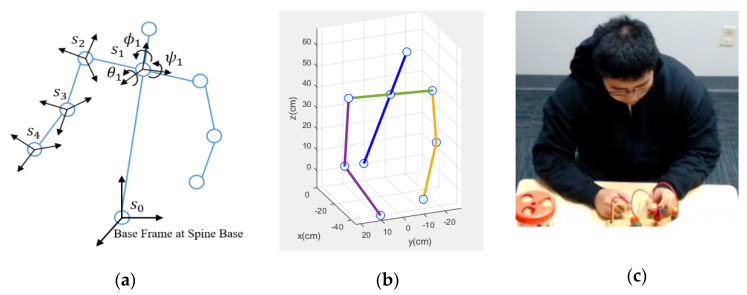
(**a**) Forward Kinematics of WINGS; (**b**) WINGS Skeleton Visualization; and (**c**) User Gesture.

**Figure 6 sensors-21-00370-f006:**
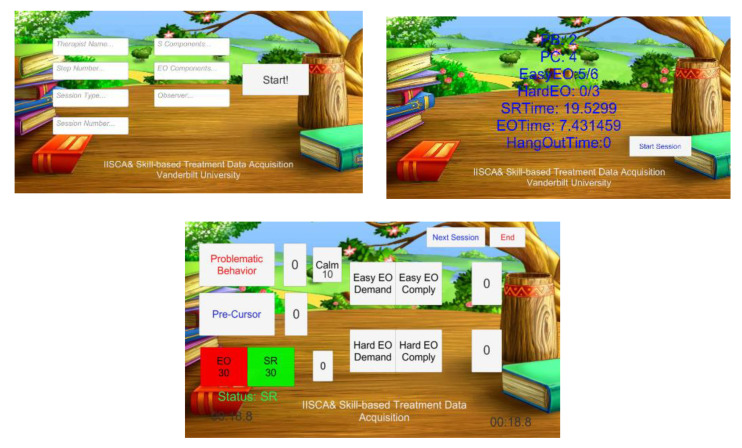
Screenshots of the Behavior Data Collection Integrator (BDCI) application.

**Figure 7 sensors-21-00370-f007:**
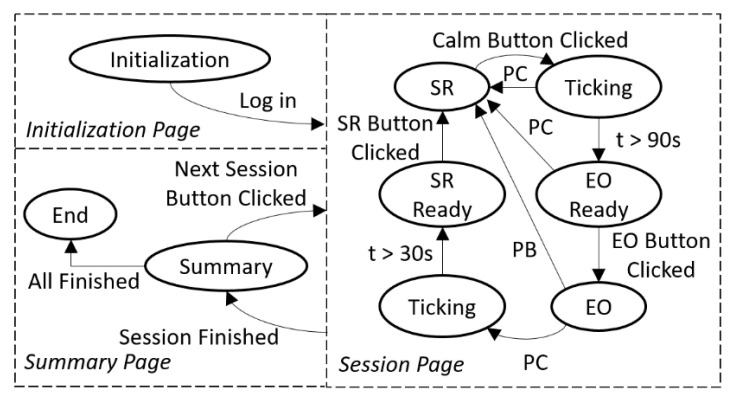
Finite state machine (FSM) of BDCI.

**Figure 8 sensors-21-00370-f008:**
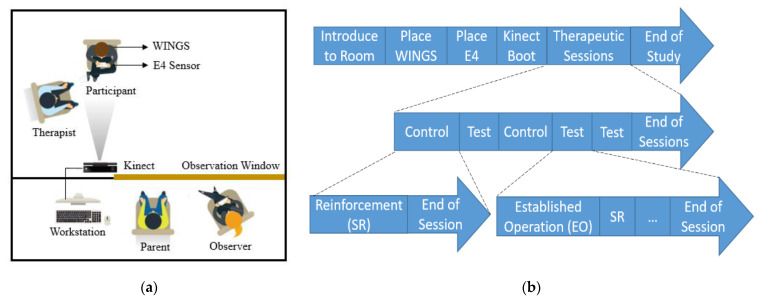
(**a**) Experimental setup; and (**b**) Experimental procedure.

**Figure 9 sensors-21-00370-f009:**
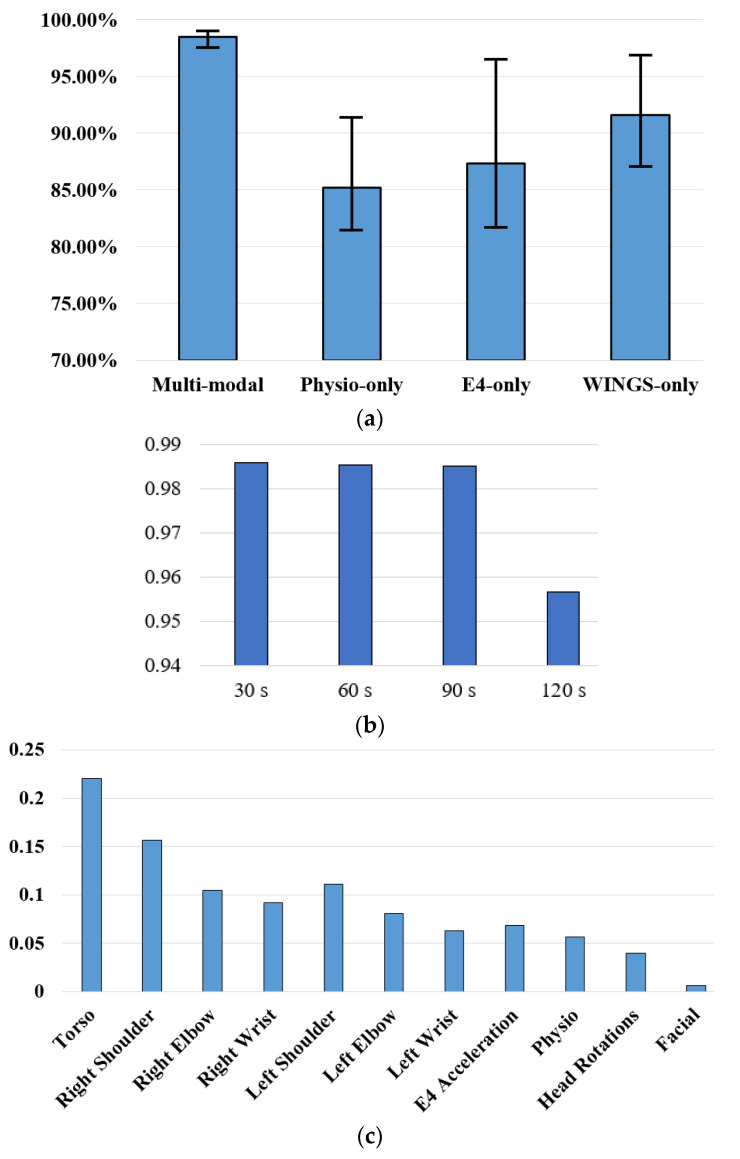
(**a**) Prediction Accuracies of Different Data Modality Models; (**b**) Interval Analysis; and (**c**) Feature Importance Comparison.

**Table 1 sensors-21-00370-t001:** Comparison of machine learning algorithms.

Machine Learning Algorithm	Individualized Model	Group Model
Random Forest	98.51%	82.36%
Support Vector Machine	88.71%	67.93%
k Nearest Neighbors	94.94%	64.36%
Decision Tree	94.76%	74.84%
Discriminant Analysis	74.55%	51.66%
Naïve Bayes	68.69%	56.57%
Neural Network	91.72%	80.17%

**Table 2 sensors-21-00370-t002:** Individual model confusion matrix.

*n* = 6004	Predicted Yes	Predicted No
Actual Yes	TP = 2365	FP = 49
Actual No	FN = 42	TN = 3548

**Table 3 sensors-21-00370-t003:** Group model confusion matrix.

*n* = 35,144	Predicted Yes	Predicted No
Actual Yes	TP = 10,993	FP = 4109
Actual No	FN = 2091	TN = 17,951

## Data Availability

The data presented in this study are available on request from the corresponding author. The data are not publicly available due to the privacy of our participants and the requirements of our IRB.

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
