# Peer review of "A Predictive Multimodal Framework to Alert Caregivers of Problem Behaviors for Children with ASD (PreMAC)"

_sensors, 2021, doi:10.3390/s21020370_

Round 1
Reviewer 1 Report
Thank you for giving me this opportunity to review the manuscript titled, “A Predictive Multimodal Framework to Alert Caregivers of Problem Behaviors for Children with ASD (PreMac)”. I am assuming that I have been selected as a reviewer because of my expertise in applied behavior analysis and, more specifically, the IISCA model that was the focal point of the manuscript. In the interest in transparency I must say that I know very little regarding the technological and computational aspects of the authors’ work. Although intriguing, my critique is entirely informed by that which I can evaluate.
The general concept of establishing a safer environment for children during behavioral assessment of problem behavior is not necessarily a new idea and the IISCA is a particular assessment model that attempts to evoke minimal problem behavior (precursors) while maintaining efficacy in informing subsequent treatment. Much like an allergy test, it requires a controlled but minimal reaction to the environmental variables assumed to be influencing the potential problem. The physician doesn’t want to cause a full blown allergic reaction, and we don’t want to cause a full blown burst in severe problem behavior. However, the authors are attempting to take the process one step further using a multimodal framework to predict when precursors to problem behavior occur. The authors seem to be asking a very socially relevant question: What if we don’t need to see problem behavior (precursors or otherwise) during an assessment at all? The results of their study support the potential for creating a safer and probably more preferred environment. For that reason I am recommending the acceptance of this manuscript. Although, I do have some minor suggestions.
The authors may want to consider delving into a very rudimentary cost-benefit analysis in their discussion. They often suggest that we must rely on paper and pencil as if it is a limitation but anyone in any country can conduct an IISCA with paper and pencil. The technology they are suggesting will only be available to a select few who would be able to afford it. Therefore, the authors may want to consider discussing if humans can also accurately predict precursors to problem behavior without the device. My assumption is that the device is more accurate, which suggests the utility over more cost efficient procedures. There is also some training that goes into collecting observational data of behavior and the necessity of including multiple observers to collect, what we term, interobserver agreement to ensure that the data collected is reliable. All of these costs could potentially be eliminated if we were to use an automated data collection device.
The authors point out the potential implication of improving safety during the assessment period but there are other strengths of this predictive model. First, it could be more preferred by the individual and their caregivers because it avoids arranging a palpably aversive environment that evokes problem behavior and the caregivers then don’t have to watch their child “break down” so to speak. The authors should discuss this in terms of social validity and how they could potentially capture this preference using questionnaires. Second, being able to predict problem behavior could also better prepare clinicians for what is to come during treatment. We have safety equipment that is worn (e.g., arm guards, helmets) to protect service providers when severe escalation occurs. The options currently are to wear that uncomfortable equipment all the time or after the burst starts to occur when it is possibly too late. If clinicians had an indication of impeding escalation, they could prepare service providers with the safety equipment before any risk to injury. This would also reduce any social stigmatization of having to be followed around all the time by a service member completely decked out in safety gear.
I also had some minor suggestions:
- Page 2, Line 53: change the sentence to read, “In addition, while the significant resources invested in an FA may result in the identification of certain environmental variables likely to contribute to problem behavior,…”
- Page 2, Line 63: precursors and problem behaviors are measured during the IISCA. We include precursors to potentially improve safety but there is no guarantee that the precursors will occur.
- Page 9, Lines 340-341: I would take out point #2. The IISCA isn’t purposefully trying to reinforce problem behavior as if to teach the child to “learn to emit the behavior with high frequency”. It arranges putative reinforcers that are said to historically influence problem behavior. Again, like the allergy example the reaction occurs because the problem exists. It is not as if the allergy test is purposefully causing the body to have a new allergic reaction.
- Page 11, Lines 414-415: the experimental design used is a multelement design not an ABABB reversal design.
- Page, 11, Line 443: The authors may want to specific what an “experimental session” is. Were they referring to how long it took to conduct the IISCA? The entire therapeutic visit? We also often report the duration of each session and the duration of the analysis (number of sessions multiplied by the session duration).
Reviewer 2 Report
The scope of the work presented, children diagnosed with ASD, is especially relevant, taking into account the difficulty in working and predicting the behaviors of these children.
The contextualization of the work, as well as the state of the art, are very adequate with a high degree of detail, adequately delving into the subject.
The presentation of the work done, as well as the results obtained, is good, making reading and comprehension easy within the complexity of the subject.
If they should improve the presentation of some figures. The setup in Figure 4.b should look more professional. Figure 6 is excessively small.
Finally, my congratulations to the authors for the work done and the presentation of it.
Reviewer 3 Report
The proposed machine learning-based predictive framework PreMAC is a original and significant work for automatically alerting caregivers of problem behaviors for children with ASD. It is a multidisciplinary work dealing with healthcare, wearable systems, machine learning, signal processing and affective computing. The proposed framework can be extended to be used in other scenarios on emotional and psychological prediction. However, the current paper should be further improved in its form and content before publication. The detailed remarks are given below.
1) The introduction is too lengthy. It should be shortened in order to be more concentrated on the main contributions of the work. The analysis on the context can be reduced. The same for the conclusion.
2) The framework and experiements as well as hardware have been clearly presented. However, the algorithms for feature selection and PreMAC training are relatively weak and too descriptive. The selected features and related methods should be clearly indicated with quantitative analysis.
3) Also, the selected features and learning data are more related to personal behaviors (face expressions, physiological data, ...) and probably not available for predicting problem behavior of a new patient. Even if the difference between indivisuliazed and group models is slight in experiments, more analysis on robustness and tolerance could be done and introduced in the prediction model. It sems that the problem behavior prediction is more related to uncertaities and imprecisions and then more relevant appraoches, shch as fuzzy logic or possibilistic approach, could be introduced. Anyway, more analysis on methodology design related to the specific scenario should be given.
4) The authors found that RF is the best method in terms of prediction accuracy. The authors should qualitatively explain why the nature of RF is more relevant to this scenario.
